# Functional Potential of Sweet Cherry Cultivars Grown in New Zealand: Effects of Processing on Nutritional and Bioactive Properties

**DOI:** 10.3390/foods14213749

**Published:** 2025-10-31

**Authors:** Ali Rashidinejad, Fatema Ahmmed, Carolyn Lister, Halina Stoklosinski

**Affiliations:** 1Riddet Institute, Massey University, Private Bag 11222, Palmerston North 4442, New Zealand; f.ahmmed@massey.ac.nz; 2The New Zealand Institute for Plant and Food Research Limited, Private Bag 4704, Christchurch 8140, New Zealand; carolyn.lister@plantandfood.co.nz; 3The New Zealand Institute for Plant and Food Research Limited, Private Bag 11600, Palmerston North 4442, New Zealand; halina.stoklosinski@plantandfood.co.nz

**Keywords:** New Zealand cherries, nutritional profiling, bioactive metabolites, polyphenols, anthocyanins, chlorogenic acid

## Abstract

**Highlights:**

**Abstract:**

While sweet cherries (*Prunus avium* L.) are globally recognized for their numerous potential health benefits, yet limited data exist on New Zealand-grown cultivars. This study examined the nutritional and bioactive profiles of six commercial cultivars—Kordia^®^, ‘Lapins’, Sweetheart^®^, Staccato^®^, ‘Bing’, and ‘Rainier’—in both fresh and processed (washed and packaged) forms. All cultivars contained notable levels of minerals, phenolics, and essential nutrients. Fresh cherries had higher mineral content (0.3–0.5 g/100 g) than processed ones (0.2–0.3 g/100 g). Carbohydrates ranged from 16.8 to 18.6 g/100 g in fresh and 15.1–17.5 g/100 g in processed cherries. Dietary fiber was slightly higher in processed samples (0.5–0.6 g/100 g) than fresh (0.2–0.5 g/100 g). Potassium, calcium, and phosphorus were more concentrated in fresh cherries. Major phenolic metabolites included neochlorogenic acid (up to 44.26 mg/100 g), (-)-epicatechin (7.89 mg/100 g), quercetin 3-rutinoside (4.34 mg/100 g), and cyanidin 3-rutinoside (80.42 mg/100 g). Processed ‘Lapins’ and ‘Bing’ retained high levels of neochlorogenic acid (40.98 and 44.26 mg/100 g), indicating minimal loss during processing. This study offers insights into the nutritional and bioactive composition of New Zealand-grown cherries, emphasizing their dietary value and health-promoting compounds such as polyphenols.

## 1. Introduction

Sweet cherries, belonging to the *Rosaceae* family and the genus *Prunus*, exhibit diverse characteristics influenced by their taxonomy and pomology, which include over a hundred species classified into the subgenera *Cerasus* and *Padus*. Pomologically, cherries are categorized based on their inflorescence type, either corymb or racemosa. These taxonomic and pomological classifications provide a scientific framework for understanding the characteristics and variations observed in the *Prunus* genus. The quality of cherry fruits is determined by various factors, including color/brightness, flavor, aroma, texture, sweetness, sourness, and texture/firmness. Beyond their visual appeal, cherries potentially offer a plethora of nutrients that contribute to various human health benefits [1,2,3].

Sweet cherries are reported in the literature to contain vitamins (A, B, C, E, and K), minerals (calcium, magnesium, potassium, and phosphorus), and carotenoids (*beta*-carotene, lutein, and zeaxanthin). Additionally, cherries contain specific polyphenols such as hydroxycinnamic acids (e.g., 3-*p*-coumaroyl quinic acid, chlorogenic acid, and neochlorogenic acid), flavanols (e.g., catechin and epicatechin), flavonols (e.g., quercetin 3-glucoside, quercetin 3-rutinoside, and kaempferol 3-rutinoside), and anthocyanins (e.g., cyanidin-3-glucoside, cyanidin-3-rutinoside, and peonidin-3-rutinoside). Cherries are also rich in carbohydrates, e.g., sugars and fiber [4,5], and organic acids such as malic acid [5]. These constituents, in particular phenolic compounds, have been associated with improvements in various health aspects, including potential protective effects against cancer [6], cardiovascular diseases [6], diabetes [7,8], and other inflammatory conditions [9,10]. Additionally, numerous studies have indicated that cherries possess immune-boosting properties by mitigating oxidative stress. For instance, cherries have been shown to improve cardio-metabolic health markers, such as lowering blood pressure, glucose, and cholesterol levels in individuals with metabolic syndrome [11]. Furthermore, cherry anthocyanins help reduce fat accumulation and oxidative stress in the liver, which can prevent the onset of non-alcoholic fatty liver disease [10]. Several studies have demonstrated that cherries contain substantial amounts of antioxidants in vitro and may exert a protective role against oxidative stress and free radical damage [12,13].

The New Zealand cherry industry has seen significant growth over the past decade, driven by rising exports to key markets like Taiwan, China, and Viet Nam, along with favorable growing conditions in the Central Otago region. In 2024, the total export value of summerfruit was close to NZ $83 million, where cherries were responsible for more than half of the total export value in this year [14]. As a result, New Zealand has been able to tap into the global demand for high-quality cherries, bolstering its export income. Over the past few years, cherry production in New Zealand has been steadily increasing, with orchards expanding and new plantings taking place. The current production and export income of cherries in New Zealand include the total land area dedicated to cherry production, which is approximately 726 hectares. The average yield of cherries in New Zealand is about 10 tonnes per hectare. The peak season for cherry harvesting occurs from December to February [14].

While cherries have been extensively studied in various regions worldwide [3,15,16], a notable research gap exists when it comes to the specific characterization of the cherries grown in New Zealand. Given the unique climatic and soil conditions, as well as the cherry cultivars cultivated in New Zealand, there is a compelling need for a deeper understanding of the phytochemical composition, potential health benefits, and market opportunities associated with these locally grown cherries. Accordingly, this research gap presents an intriguing opportunity to explore the distinctiveness of New Zealand cherries and unlock their full potential, both in terms of health and economic value. Thus, the current research aimed to address such a gap by conducting a comprehensive analysis of six main cherry cultivars cultivated in the cherry orchards situated within the Otago region of the South Island, New Zealand.

We comprehensively investigated the nutritional components of New Zealand-grown cherry cultivars, including sugar profiles, vitamins, minerals, and amino acids. We also explored bioactive properties such as total phenolic content (TPC) and quantified individual phenolic metabolites, including polyphenols and anthocyanins.

## 2. Materials and Methods

### 2.1. Sample Harvest, Processing, and Preparation

Six cherry cultivars (Kordia^®^, ‘Lapins’, Sweetheart^®^, Staccato^®^, ‘Bing’, and ‘Rainier’) were harvested (at the same degree of ripeness) randomly by hand-picking from an orchard in the Otago region of the South Island in New Zealand and labeled according to their sampling regime: ‘fresh’ or ‘processed’. The samples were in two forms: fresh-frozen cherries (i.e., freshly picked and then frozen at the orchard as soon as possible; referred to as “fresh” in this paper) and cherries that were sent to the processing facility and underwent washing and freezing at the commercial scale (referred to as “processed” in this paper). In this paper, the term “commercial processing” refers to the standard procedures routinely employed in the processing lines of cherry processing in New Zealand. The cleaning of cherries was performed using standard water-based washing.

The six cherry cultivars were selected because they are commercially significant and among the most commonly grown varieties in the Otago region of the South Island, New Zealand. They represent the primary cultivars supplied to both domestic and export markets, making them relevant for assessing differences between fresh and processed cherries. The cherries were harvested in three replications, with samples analyzed in triplicate unless stated otherwise. The collection period spanned from December 2021 to mid-February 2022. After harvesting, 1 kg of cherries was promptly de-pitted using a pitter apparatus and immediately frozen on-site to preserve freshness and nutritional properties, labeled as ‘fresh’. Another 1 kg of cherries from each cultivar was de-pitted and packaged at a specialized facility, using suitable packaging materials. These commercially processed cherries were then frozen in the same manner to ensure sustained quality and freshness, and were labeled as ‘processed’. The processed cherry samples were packaged under normal atmospheric conditions, without vacuum or modified-atmosphere packaging. They were transferred to the laboratory immediately after packaging and maintained at −20 °C throughout transport and storage, minimizing any potential changes in composition or quality due to the packaging method.

The pH of both fresh and processed cherry samples was determined using a calibrated pH meter (Model HI 221, Hanna Instruments Ltd., Bedfordshire, UK) after preparing a homogenized slurry with an appropriate amount of distilled water. Upon arrival, the cherries were stored in a freezer at −20 °C and subsequently freeze-dried using a Cuddon FD18 Freeze Dryer (Cuddon Freeze Dry, Blenheim, New Zealand). The resulting dried flesh was ground, packed in airtight containers, and stored at −20 °C until further analyses. The team at Plant and Food Research (Palmerston North, New Zealand) also received a subsample of cherries for polyphenol analysis. These were freeze-dried using an Operon freeze dryer (Operon; FDTE-5025; Gimpo-si, Korea) and subsequently ground and stored as above. Although different freeze-dryer models (Cuddon and Operon) were used, the operating conditions were standardized—including temperature (−50 °C condenser, 0.1 mbar pressure) and drying duration. Both instruments were calibrated and operated under comparable parameters to ensure consistent moisture removal and sample integrity. Therefore, any differences in compound extraction due to equipment type are expected to be negligible.

### 2.2. Chemicals

Methanol, Folin-Ciocâlteu reagent, gallic acid, HCl, HNO3, β-d-glucopyranoside, isooctane, hydroxylamine hydrochloride, pyridine, hexamethyldisilazane, trifluoroacetic acid, and Trolox (6-hydroxy-2,5,8-tetramethylchroman-2-carboxylic acid), were all purchased from Sigma-Aldrich (St. Louis, MO, USA). To ensure the highest purity, deionized water was obtained from a Milli-Q Ultra-pure water system (Millipore, Billerica, MA, USA). All chemicals and reagents were of analytical grade and used as received without any additional purification, ensuring consistency and accuracy in the experimental procedures.

### 2.3. Proximate Composition

The proximate compositions of the six cherry cultivars were analyzed using standard AOAC methods (AOAC, 2010) [17]. The moisture content was determined through the oven-dried method (AOAC 925.10/930.15), involving the weighing of the sample before and after drying it in an oven at 105 °C for 12 h to ascertain the loss of water content [17].

The ash content was determined by subjecting the sample to combustion at 550 °C for 5–6 h, utilizing a muffle furnace according to the AOAC 942.05 method [17]. The protein content was assessed in accordance with AOAC 968.06, while the fat content was determined using AOAC 954.02 methods [17]. The carbohydrate content was calculated by subtracting the sum of moisture, protein, lipid, total dietary fiber, and ash from 100, as expressed in the following equation:*Total available carbohydrate* (%) = 100 − (*moisture* + *protein* + *lipid* + *total dietary fiber* + *ash*)(1)

The total fiber content, including insoluble and soluble dietary fiber, was calculated following the standard AOAC method (AOAC 991.43) [17]. This method is widely used for the determination of dietary fibre content in food.

### 2.4. Amino Acid Profile

The amino acid composition (histidine, isoleucine, leucine, lysine, methionine, phenylalanine, threonine, tryptophan, and valine) was analyzed using high-performance liquid chromatography (HPLC; (Thermo Fisher Scientific, Waltham, MA, USA)) following the standard AOAC method (AOAC 994.12) [17]. The samples underwent hydrolysis with hydrochloric acid, and the resulting hydrolysate was separated by HPLC using a reversed-phase column. A fluorescence detector was employed to detect amino acids. The quantitative results of the amino acid analysis were expressed in milligrams (mg) of amino acid per gram (g) of the sample.

### 2.5. Sugars

The sugar profile of the six cherry samples was determined using gas chromatography coupled with a flame ionization detector (GC-FID; Agilent Solution, Inc., Santa Clara, NY, USA). The analysis included the quantification of fructose, glucose, lactose anhydrous, maltose, sucrose, and galactose present in the fruit samples. To prepare the samples, 0.6 mg of freeze-dried powder of each cherry sample was taken in a test tube and 0.5 mg of the internal standard (β-d-glucopyranoside) was added. The test tubes were incubated at 70 °C for 5 min using a water bath, with 12.5 mg of hydroxylamine hydrochloride and 0.5 mL of 99% pyridine (Sigma-Aldrich). After cooling for five minutes, 0.5 mL of 99% hexamethyldisilazane and 0.4 mL of 98% trifluoroacetic acid (Sigma-Aldrich) were added to the test tubes. The mixture was continuously shaken and allowed to react at room temperature (23 °C) for 10 min. Following this, 0.5 mL of undiluted isooctane and 4 mL of de-ionized water were added to achieve phase separation. The top isooctane layer was transferred to GC vials for analysis.

### 2.6. Vitamins

Vitamins A and E were determined according to modified AOAC methods (992.06 and 992.03) [17]. Vitamin A value was expressed in retinol equivalents per 100 g of sample, whereas vitamin E was quantified as alpha tocopherol equivalents. Vitamin C was determined using HPLC, according to the method described by [18]. The B vitamins (B1, B2, B3, and B6) were determined using the modified [17,19] method reported in previous studies [20,21]. The quantitative determination of vitamins was expressed in milligrams (mg) of vitamin per 100 g of the sample.

### 2.7. Minerals

Inductively coupled plasma optical emission spectroscopy (ICP-OES) was used to determine mineral content. Different concentrations (10–50 mg/L) of ICP multi-element standard solution containing a mixture of minerals (Na, K, Mg, P, Zn, Se, Fe, Mn, I, Ca, Cu, Zn) were used as calibration solutions. 1% HNO_3_ was used as a diluent to achieve the desired concentration of the standard solution. The quantitative determination of minerals was expressed in milligrams (mg) of minerals per 100 g of the sample.

### 2.8. Total Phenolic Content

The TPC values of the cherry samples were assessed using a modified Folin-Ciocâlteu assay [22] suitable for microplate readers. Optical density measurements at a wavelength of 760 nm were performed using a bench-top microplate reader (BioTek, Winooski, VT, USA). Various dilutions of gallic acid (ranging from 0 to 1000 µg/mL) were prepared to develop a standard curve. The obtained data were expressed as mg gallic acid equivalents (GAE)/g of ground freeze-dried cherry powder.

### 2.9. Phenolic Metabolites

Phenolic metabolites were measured in the fresh and processed cherry samples as polyphenols or anthocyanins. ‘Bing’ and ‘Rainer’ fresh samples were not available for analysis due to the seasonality limitation and shortage of the samples. A detailed list of individual metabolites measured in this investigation can be found in Appendix A. One replicate of freeze-dried cherry sample was solvent-extracted overnight and diluted appropriately with methanol for polyphenol and anthocyanin analysis, as detailed below. The quantitative determination of phenolic metabolites was calculated back to fresh weight of the cherry, and expressed in micrograms (µg) of phenolic metabolite per gram (g) of the sample.

#### 2.9.1. Phenolic Compounds (Non Anthocyanins)

Polyphenols were measured using ultra-high-pressure liquid chromatography–high-resolution time-of-flight mass spectrometry (UHPLC-HR-TOF-MS). The polyphenols were separated on a Thermo Scientific Vanquish UHPLC system using a standard reversed-phase column with a solvent gradient of 0.2% formic acid and acetonitrile. The UHPLC system was connected to a timsTOF mass spectrometer with an electrospray ion source (Bruker Daltonik, Bremen, Germany). Negative ion electrospray was used with a capillary voltage of 3500 V. Polyphenols were detected in standard MS mode, and Exact Ion Chromatograms (EIC) were generated for the accurate mass of each target metabolite. The identity of each metabolite was confirmed based on accurate mass and LC retention time. Data were processed using TASQ 2022 (Bruker Daltonics, Bremen, Germany), and polyphenolic concentrations were calculated by batch comparison with external calibration curves of authentic compounds. Where authentic compounds were unavailable, surrogate quantitation was performed, choosing an authentic compound structurally similar and with known response factors.

#### 2.9.2. Anthocyanins

Anthocyanins were measured using a Thermo Dionex Ultimate 3000 HPLC (Thermo Fisher Scientific, Waltham, MA, USA). Separation of the anthocyanins was achieved using a standard reversed-phase column with a solvent gradient composed of 1% phosphoric acid and acetonitrile. Anthocyanins were detected using a diode array detector at 530 nm and the spectra data (250–600 nm) were collected for metabolite confirmation. The chromatographic data system was Chromeleon 7 (Thermo Fisher Scientific, Waltham, MA, USA), and all anthocyanin concentrations were calculated as cyanidin 3-glucoside equivalents using a cyanidin 3-glucoside calibration curve.

### 2.10. Statistical Analysis

All tests (except for polyphenols and anthocyanins) were performed in triplicate. Data are expressed as mean ± standard deviation. One-way analysis of variance (ANOVA) and post hoc Tukey’s multiple comparisons tests were performed to analyze the experimental data using Minitab software (Version 16.1, Minitab Limited, Sydney, Australia). The differences were considered statistically significant at *p* < 0.05.

## 3. Results and Discussion

The research evaluated the nutritional and bioactive properties of cherries in two forms: “fresh” (immediately frozen at the orchard) and “processed” (subjected to washing, de-pitting, and packaging before freezing). The comprehensive analysis included key compositional parameters such as dry matter, macronutrients, dietary fiber, sugar profiles, vitamins, minerals, and bioactive metabolites across six cultivars: Kordia, Sweetheart, Staccato, ‘Lapins’, ‘Bing’, and ‘Rainier’. The findings for each of these analyses are presented on fresh weight (FW) basis and discussed below.

### 3.1. Dry Matter Content

As shown in Figure 1, fresh cherries consistently exhibited higher dry matter (DM) content than processed cherries across all cultivars, probably because of water loss during processing. Fresh Staccato cherries showed the highest DM content (21.4%), while processed Sweetheart cherries were highest among processed samples (20.2%).

In general, cherries tend to lose water during processing and packaging [23,24]. However, in the case of the current study, there was no significant difference (*p* > 0.05) in terms of moisture content between processed and fresh samples of the same cultivar. A previous study [25] found that the moisture content of sweet cherries exceeded 85%, which is slightly higher than the values observed in the current study. Two sweet cherry cultivars, ‘Bing’ and ‘Dawson’, were reported to contain 77% and 83.1% moisture content, respectively [26]. Both ‘Della Recca’ and ‘Del Monte’ *Prunus avium* cherries were reported to have a similar moisture content, approximately 83% [2]. The moisture values (78.6–82.0%) of the six cherry cultivars in this study are consistent with these previous findings. All samples were handled under standardized postharvest conditions, with comparable harvest-to-freezing intervals, identical packaging, and storage at −20 °C, thereby minimizing variability from external factors. Consequently, differences in water loss are primarily attributed to processing effects.

### 3.2. Chemical Composition

Overall, based on the data presented in Table 1, we found low amount of ash content (0.2–0.5 g/100 g) in both fresh and processed cherries. However, we did not find any significant variation (*p* > 0.05) in ash content between fresh and processed samples, suggesting that the processing (i.e., washing and packaging) did not significantly affect the mineral content of the samples. Kordia had a significantly higher (*p* < 0.05) ash content than either the fresh or processed cherries of any other cultivars in our study.

Like other fruits, cherries are generally low in fat (0.5–0.8 g/100 g), protein (0.7–1.1 g/100 g), and dietary fiber (0.8–1.3 g/100 g) [27]. This study did not find any significant differences in the fat content between fresh and processed cherry samples. The total fat content observed in this study aligns with data previously reported by the New Zealand Food Composition Database (NZFCD), which indicated fat concentrations ranging from 0.23 g/100 g to 0.74 g/100 g in both fresh and frozen cherries (NZFCD) [28]. Specifically, fresh cherries contained approximately 0.21 g/100 g fat, while this value for processed cherries was around 0.25 g/100 g.

The protein content of all cherry samples in this study was reasonably low. Although fresh Sweetheart contained relatively higher protein content (1.1 g/100 g) than the other cultivars, at such low concentrations (Table 1), the differences are not of nutritional significance. The protein content observed in different cultivars of cherries in the current study (0.7–1.1 g/100 g) is in line with that reported in previous research, which found protein concentrations in sweet cherries within the range of 0.8 g/100 g to 1.4 g/100 g [29].

The total available carbohydrate content measured in this study was the predominant biochemical constituent (15.1–18.6 g/100 g), with the total dietary fiber at 0.8–1.3 g/100 g (Table 1). Carbohydrates encompass dietary fiber and various sugars, including fructose and glucose, which contribute to the sweet and distinctive flavor of cherries, making them a popular fruit choice. The results obtained in the present study are in agreement with those reported in some previous research carried out on some cherry cultivars in other parts of the globe [1,28]; where carbohydrates were reported to make up around 12 g/100 g to 17 g/100 g of the fruit’s composition, with total dietary fibers ranging from 1.3 g/100 g to 2.1 g/100 g. The carbohydrate content in fresh Staccato (18.6 g/100 g) was relatively higher than those in other fresh and processed cherry cultivars (Table 1). Fresh Kordia contained more dietary fiber than other fresh cherry samples. Nevertheless, the total available carbohydrate content found in the cherry cultivars under study did not reach dietary significance (see Appendix A). While specific data for cherries are not available in the NZFCD, the observed values are consistent with the upper range of carbohydrate content typically found in sweet fruits consumed in New Zealand and countries alike [28].

#### 3.2.1. Vitamins

Most vitamins including A, B1, B2, B3, B6, B9, C, E, and K were detected and quantified in both fresh and processed cherry samples (Table 1). The results showed that these specific cultivars of cherries grown in New Zealand are sources of ascorbic acid (vitamin C), which is reported to exhibit a strong antioxidative effect [30]. The vitamin C content in cherries in this study ranged from 4.13 mg/100 g to 5.70 mg/100 g, which is slightly lower than the concentrations in sweet cherry cultivars (around 7.0 mg/100 g) reported previously [30]. Other studies [29,31] reported that vitamin C contents in sweet and sour cherry cultivars varied from 4.0 to 10.0 mg/100 g fresh weight. This variation can be attributed to differences in cherry species and cultivars, as well as different maturities at harvest and growing/environmental conditions [29]. On the other hand, the vitamin C contents in some other stonefruits, such as apricots (4 mg/100 g), nectarines (10 mg/100 g), peaches (3 mg/100 g), and plums (6 mg/100 g) were reported to be within a similar range to those of the cherry cultivars studied here [32]. Another study reported a wide range of vitamin C content (1–14 mg/100 g) in seven peach cultivars. This finding supports the hypothesis that the observed vitamin C content in cherries aligns with broader patterns seen in other fruits.

In the current research, we found no significant variation (*p* > 0.05) in vitamin C content among the fresh and processed cherry samples, regardless of cultivar (Table 1). For vitamins B1, B2, and B6, the concentrations remained relatively stable between the fresh and processed samples for all six cherry cultivars (Table 1). Based on the literature, the concentration of vitamins B1, B2, and B6 can range from 0.02 to 0.05 mg/100 g in various cultivars [29], which aligns with our findings here (0.02–0.09 mg/100 g). According to the NZFCD, typical concentrations in similar food products are approximately 0.03 mg/100 g for B1, 0.04 mg/100 g for vitamin B2, and 0.05 mg/100 g for vitamin B6, further supporting our results [28]. The concentration of vitamin B3 was slightly lower (0.03–0.15 mg/100 g) than in some previous reports; e.g., 0.15–0.40 mg/100 g reported by Serradilla et al. [29], and slightly below the NZFCD average of 0.20 mg/100 g [28]. Furthermore, the concentration of vitamin E in Kordia and Staccato cherries was significantly higher (*p* < 0.05) than those in the other cherry cultivars (Table 1). The vitamin E content in our studied cherry cultivars ranged from 0.1 to 0.2 mg/100 g, which is slightly lower than that of apples (0.2–0.5 mg/100 g) and strawberries (0.3–0.4 mg/100 g) according to Bohn and Bouayed [27]. Vitamin K was the next most abundant vitamin in our samples. Notably, processed Kordia had a relatively higher concentration of vitamin K (3.99 μg/100 g) than the other cherry cultivars (Table 1). The observed variation in vitamin K content can be linked to the differences in cherry species and cultivars, as reported earlier [33]. Even though fruits are not generally recognized as important sources of vitamin K (Appendix A), some cultivars of sweet cherries may contain higher amounts of this vitamin than many other commonly consumed fruits. For instance, apples (‘Fuji’: 0.9–1.1 μg/100 g; ‘Gala’: 1.2–1.5 μg/100 g), bananas (0.5–1.0 μg/100 g), strawberries (1.6–4.1 μg/100 g), watermelon (0.1–0.1 μg/100 g), nectarines (1.3–2.8 μg/100 g), oranges (0.1–0.1 μg/100 g), and pineapples (0.2–1.2 μg/100 g) all contain less vitamin K than cherries [34]. Lastly, our findings also align with the vitamin K content reported in various other cherry cultivars (1.8–4.0 μg/100 g) [33].

#### 3.2.2. Minerals

As presented in Table 1, twelve minerals including calcium, magnesium, potassium, sodium, phosphorus, iron, copper, iodine, manganese, selenium, zinc, and chloride were detected and quantified in six cultivars of fresh and processed cherry samples. Potassium was the most abundant mineral across all cultivars of both fresh and processed cherries, followed by phosphorus, calcium, and magnesium (Table 1). All six cultivars of fresh cherries displayed higher potassium concentrations (around 220 mg/100 g) than their processed counterparts (200 mg/100 g). This variance could potentially be attributed to the washing process during processing, which might leach out some of the potassium content present in cherry cultivars. Notably, fresh Kordia, Sweetheart, and Staccato exhibited even higher potassium concentrations (220 mg/100 g) than the fresh ‘Lapins’ fruit (184 mg/100 g). Additionally, processed Kordia recorded the highest potassium content (192 mg/100 g), of all the processed cherry cultivars.

Calcium content was in the range of 11.3–14.8 mg/100 g for fresh samples and 12.2–14.2 mg/100 g for processed samples, with the highest calcium content consistently found in both fresh and processed Staccato (Table 1). Fresh Sweetheart (12.5 mg/100 g) and processed ‘Rainier’ (12.8 mg/100 g) displayed higher magnesium content than other cherry cultivars (Table 1). Phosphorus content also notably stood out in fresh Sweetheart (25.0 mg/100 g) compared with other cherry cultivars (Table 1). The phosphorus concentration was identical (21.0 mg/100 g) in processed Kordia, Sweetheart, and ‘Rainier’ fruit. Fresh Sweetheart cherries (0.28 mg/100 g) and processed ‘Bing’ fruit (0.23 mg/100 g) displayed elevated concentrations of iron compared with other cultivars.

Other trace minerals, including copper, iodine, manganese, selenium, and zinc, were present at very low concentrations (<0.2 mg/100 g) in the studied cherry cultivars (Table 1). Remarkably, processed samples (0.058–0.119 mg/100 g) exhibited slightly higher manganese concentrations than fresh cherry samples (0.051–0.072 mg/100 g). Therefore, it can be said that processing influenced the mineral composition of cherries in a non-uniform manner, reflecting the distinct chemical behavior of individual elements. The reduction in potassium content observed after processing is likely due to its high solubility and mobility, which make it prone to leaching during washing or drying steps. In contrast, minerals such as calcium and magnesium, which often form stable complexes with cell wall components or organic acids, may be less susceptible to such losses. The apparent increases in some trace elements may result from concentration effects associated with moisture removal during freeze-drying or improved extractability due to matrix disruption.

These current results confirm that sweet cherries contain essential minerals, including potassium, phosphorus, calcium, and magnesium. These findings align with previous studies that have highlighted cherries as sources of these specific minerals [9,10,29]. This indicates consistency in the major mineral composition between our processed cherries and their fresh counterparts. While potassium, calcium, magnesium and phosphorus, dominated the mineral profile, it is worth noting that calcium (11.6–14.8 mg/100 g), magnesium (9.5–12.0 mg/100 g), and phosphorus (18.3–25.0 mg/100 g) were also present in concentrations consistent with those in a previously reported study [29]. However, when assessing against the recommended dietary intakes for food labeling purposes outlined by Food Standards Australia New Zealand 2024 [35] for phosphorus (1000 mg/100 g), calcium (800 mg/100 g), and magnesium (320 mg/100 g), cherries do not stand out as significant sources of these minerals.

### 3.3. Sugar Profiles

Based on the findings of this research, glucose and fructose were the dominant sugars in all cherry cultivars, accounting for 90% of the total fruit sugars (Table 2). The other sugars such as sucrose and galactose could not be quantified as their concentrations were below the detection limit (Table 2). Fresh Staccato contained the highest concentration of glucose (8.0 g/100 g) among the six types of cherry samples we analyzed (Table 2). The glucose content of the samples in the current research (6.6–8.0 g/100 g) is similar to that reported for some Canadian and Hungarian cultivars, with glucose contents in the range of 5.2 g/100 g to 8.8 g/100 g [36] and 6.6 g/100 g to 9.1 g/100 g [37], respectively. However, in some Turkish cultivars, a higher glucose content (15.5–21.5 g/100 g) has been reported than those of glucose content of the currently studied cherry cultivars (6.6–8.0 g/100 g) [36].

The present study also revealed that fructose was the second most dominant sugar across all samples (Table 2), ranging from 6.2 g/100 g to 6.6 g/100 g in fresh samples and 5.7 g/100 g to 6.5 g/100 g in processed samples, depending on the cultivars. These results align with previous studies on different cherry cultivars, where fructose content was reported to be from 4.4 to 6.4 g/100 g [34] to 4.7–10.1 g/100 g [38]. Another study reported that fructose content in cherry was between 3.5 and 4 g/100 g [37]. A higher fructose content was found in fresh Sweetheart and fresh Staccato cherries than in our other cherry samples (Table 2). Processed Kordia contained higher fructose content (6.5 g/100 g), followed by other processed cherry cultivars (Table 2). The lowest fructose content (5.7 g/100 g) was determined in processed ‘Lapins’ fruit (Table 2).

In terms of total sugars, fresh samples of both Sweetheart and Staccato cherries displayed the highest content, approximately 14.5 g/100 g. The total sugar content in the processed ‘Lapins’ was around 12.2 g/100 g, which was lower than those of our other processed cherry cultivars. Thus, the available sugar content in the cherry samples studied typically ranged from 12 g/100 g to 15 g/100 g. Previous research has reported that sugar content can range from 8.0 g/100 g to 11 g/100 g, depending on the cherry cultivars [29,39]. Other studies have found that the sugar content in eleven different cherry cultivars (‘Lambert’, ‘Merton Late’, ‘Starks Gold’, ‘Van’, ‘Vista’, ‘Ziraat’, ‘Belge’, ‘Summit, and Sweetheart) falls within the range of 10 g/100 g to 15 g/100 g [40,41]. However, sugar composition can be variable even within the same cultivar [29] and depends on climatic conditions, culture systems, and rootstock [42,43,44]. As cherries ripen, their glucose, fructose, and sucrose contents increases, creating a sweetness that appeals to our taste buds. Conversely, some phenolic metabolites contribute to a dry, puckering sensation known as astringency. The sweetness in cherry cultivars primarily comes from glucose and fructose, while polyphenolic metabolites significantly enhance the sensory and organoleptic qualities of cherries, including color, taste, and astringency [45,46]. Factors such as ripeness, growing conditions, and postharvest handling affect the balance between sugars and phenolic metabolites, shaping the overall flavor profile of cherries [47].

### 3.4. Amino Acid Profile

Amino acids represent a pivotal class of nutrients for human consumption, forming the building blocks of proteins in all living organisms [48]. The amino acid profiles for all cherry samples evaluated in the current study are presented in Table 3. We identified and quantified eighteen common amino acids from the six cherry cultivars, with nine of them being classified as essential amino acids (histidine, isoleucine, leucine, lysine, methionine, phenylalanine, threonine, tryptophan, and valine). Overall, in the context of human health concentrations of amino acids are low in cherry. Aspartic acid was the most abundant amino acid found in the six cherry cultivars, with fresh Kordia (3.75 mg/g) and Sweetheart (3.61 mg/g) containing the highest concentrations. The processed samples of ‘Bing’ also contained relatively higher concentrations of aspartic acid (3.01 mg/g) than the other processed cherry cultivars.

Fresh Kordia and Sweetheart contained substantially higher glutamic acid content, measuring 0.65 and 0.62 mg/g, respectively, surpassing the concentrations found in other studied cultivars. In contrast, similar concentrations of glutamic acid were observed in all types of processed cherry cultivars (Table 3). Fresh Sweetheart and processed ‘Bing’ exhibited slightly higher proline content, measuring 0.57 mg/g and 0.44 mg/g, respectively. Despite minor variations, most amino acid concentrations remained consistent between fresh and processed cherries (Table 3). Overall, Kordia displayed significantly higher (*p* < 0.05) concentrations of the tested amino acids than other cherry cultivars, except for proline, cysteine, methionine, and tryptophan (Table 3). The findings of the current study are consistent with those reported in prior research [2,29,34].

### 3.5. Total Phenolic Content (TPC)

The TPC values, measured by the Folin-Ciocâlteu assay, serve as indicators of the antioxidant potential of plant samples, because of the strong correlation between TPC and in vitro antioxidant activity. It is notable that while the Folin-Ciocâlteu assay is widely used to estimate the TPC of plant samples, this method is not specific to polyphenols. The reagent can react with other reducing substances, such as certain vitamins, amino acids, and sugars, which can also contribute to the measured TPC. Therefore, the results obtained from this assay may overestimate the actual polyphenol content, as it includes these other compounds. This limitation should be considered when interpreting the antioxidant potential and phenolic content of the samples. Nonetheless, in the context of the current study, comparing various cherry cultivars for their TPC values ensures that the results are consistent and reliable. This approach allows us to draw meaningful interpretations and conclusions about the relative phenolic contents and antioxidant potentials of different cherry cultivars. Among the six cherry cultivars evaluated, both fresh and processed Staccato cherries exhibited the highest TPC values (1.5 mg GAE/g FW), surpassing the other cultivars (Figure 2).

Fresh cherries, specifically Kordia, ‘Lapins’, Sweetheart, and Staccato, exhibited similar TPC concentrations to their processed counterparts (Figure 2). Notably, the TPC values for these four cherry cultivars were significantly higher than those of the processed ‘Bing’ (0.6 mg gallic acid equivalents (GAE)/g FW). There were not statistically significant (*p* < 0.05) differences in TPC values between the fresh and processed samples of the same cultivar, although this may vary among different cherry cultivars. Additionally, the TPC values in the six studied cherry samples ranged from 0.6 (‘Bing’) to 1.5 mg GAE/g (Staccato). This aligns with previous research conducted by Usenik et al. [49] on 13 sweet cherry cultivars in Northern Europe, where TPC values were reported to range from 0.44 mg GAE/g (‘Lapins’) to 0.9 mg GAE/g (‘Ferprime’) [49]. Another study conducted by Ballistreri et al. [3] found varying concentrations of total phenolics in 24 cherry cultivars grown in southern Europe, ranging from 0.84 mg GAE/g (‘Napoleona Grappolo’) to 1.9 mg GAE/g (‘Sonata’) [3].

Another study measured the concentration of total phenolics in sweet cherries, which ranged from 0.92 mg GAE/g (Black Gold^®^) to 1.5 mg GAE/g (‘Heartland’) across eight different cultivars grown in New York [16]. Moreover, Tabart et al. [22] reported a wide range of TPCs in cherries, with sweet cultivars varying from 0.52 to 5.8 mg GAE/g. These results align with the findings of our current study on the sweet cherry cultivars grown in New Zealand. In contrast, sour cherries exhibited higher phenolic contents than sweet cherries.

In the case of sweet cherries, with a TPC ranging from 0.6 to 1.5 mg GAE/g, yet they provide a moderate source of antioxidants. This phenolic content surpasses that of jackfruit (0.9 mg GAE/g), avocado flesh (1.3 mg GAE/g), green-fleshed kiwifruit (1.12 mg GAE/g), and raspberries (1.26 mg GAE/g) [48,49]. However, it is slightly lower than those of fruits such as mango flesh (2.4 mg GAE/g), tamarind flesh (3.9 mg GAE/g), evergreen blackberries (5.0 mg GAE/g), orange (2.4 mg GAE/g), fig (4.6 mg GAE/g), and blueberries (5.3 mg GAE/g) [50,51,52]. While sweet cherries may not possess the highest phenolic content among common fruits, they still make a contribution to a balanced diet rich in antioxidants, which are essential for combating oxidative stress and promoting overall health.

### 3.6. Phenolic Composition

The targeted set of polyphenols to quantify was determined after the literature was reviewed and a sweet cherry composite sample was analyzed using untargeted MS/MS, with criteria for identification that included accurate mass, isotope ratios, molecular formula, LC retention time and molecular fragmentation. Sweet cherry’s three dominant anthocyanins were measured separately from the polyphenols owing to their differing chemical properties. A total of six classifications of phenolic metabolites were quantified (Table 4). These metabolites included hydroxycinnamic acids (3,5-dicaffeoylquinic acid, 3-*p*-coumaroyl quinic acid, chlorogenic acid, ferulic acid, neochlorogenic acid, *trans*-4-coumaroyl quinic acid and *trans*-5-coumaroyl quinic acid), flavanols (catechin, (-)-epicatechin), flavonols (kaempferol 3-rutinosin, quercetin 3-galactoside. quercetin 3-glucoside, quercetin 3-rhamnoside, quercetin 3-rutinoside), flavanones (sakuranetin and sakuranin), procyanidins (procyanidin B1, B2, B5 and B7), and anthocyanins (cyanidin 3-glucoside, cyanidin 3-rutinoside, and peonidin 3-rutinoside). These findings are consistent with previous studies that have identified sweet cherries containing notable amounts of health-promoting metabolites, although the specific health benefits may vary with overall dietary intake [28,49,53].

Cherries are known to contain considerable amounts of phenolic metabolites. The hydroxycinnamic acids, 3-*p*-coumaroylquinic acid, neochlorogenic acid, chlorogenic acid, and 3,5-dicaffeoylquinic acid, were the primary types of polyphenols found in sweet cherries in this study, consistent with the findings of prior research [29]. Notably, 3-*p*-coumaroylquinic acid and neochlorogenic acid were found in higher concentrations than other hydroxycinnamic acids analyzed in the cherry cultivars (Table 4).

Among the fresh cherries, Staccato contained the highest concentration of neochlorogenic acid (29.50 mg/100 g), whereas processed ‘Bing’ exhibited the highest concentration overall (44.26 mg/100 g) (*p* < 0.05). Furthermore, 3-*p*-coumaroylquinic acid was found in significantly higher concentrations (*p* < 0.05) in the processed Kordia than in the other cherry cultivars (Table 4). However, chlorogenic acid content showed no significant variation (*p* > 0.05) among processed Kordia, ‘Lapins’, Sweetheart, and Staccato. Notably, when comparing processed Kordia, ‘Lapins’, and Sweetheart with their fresh counterparts, we observed that processed cherries had a significantly higher (*p* < 0.05) chlorogenic acid content than the fresh cultivars (*p* < 0.05). The concentrations of neochlorogenic acid (ranging from 17.87 to 44.26 mg/100 g) and 3-*p*-coumaroylquinic acid (ranging from 1.95 to 34.21 mg/100 g) reported in this study are consistent with previous findings [28]. ‘Rainier’ was found to contain a lower concentration of 3-*p*-coumaroylquinic acid (1.95 mg/100 g) than other cultivars (Table 4), which is consistent with the findings of prior research [3] and is associated with the antioxidant potency of cherry extracts on human low-density lipoproteins [53].

Substantial concentrations of flavanols were found in both fresh and processed Staccato, followed by other cherry cultivars (Table 4). Processed Sweetheart, Kordia, and ‘Lapins’ cherries had significantly higher (*p* < 0.05) epicatechin content than their respective fresh counterparts. Moreover, processed ‘Rainier’ cherries exhibited significantly lower (*p* < 0.05) amounts of catechin and epicatechin than the other processed cultivars. Additionally, the flavonols quercetin 3-rutinoside and kaempferol 3-rutinoside were significantly higher (*p* < 0.05) in processed Kordia than in the other cherry cultivars. Quercetin 3-glucoside was found in higher amounts in both fresh Kordia (0.55 mg/100 g) and processed Kordia (0.73 mg/100 g), followed by other cherry cultivars (Table 4). Other flavonols, such as quercetin 3-galactoside and quercetin 3-rhamnoside, were found in low concentrations or often not detected.

Kordia also exhibited the highest concentrations of both analyzed flavanones (sakuranetin and sakuranin). The maximum concentration of sakuranetin (0.03 mg/100 g) was found in fresh Kordia, while the highest concentration of sakuranin overall (0.41 mg/100 g) was found in the processed form of the same cultivar. We did not expect to find these two metabolites in high concentration in cherry fruit; however, sakuranetin is a major metabolite in cherry stems, and in our own analysis of cherry stems, we found sakuranin to be higher again than sakuranetin (data not published). These two metabolites may be responsible for the perceived traditional medicinal health benefits of consuming cherry stems.

In terms of procyanidin dimers, including procyanidin B1, procyanidin B2, procyanidin B5, and procyanidin B7, the highest concentration of procyanidin B2 was consistently found in both fresh and processed Staccato cherries (4.16 and 3.34 mg/100 g, respectively). When comparing fresh and processed Sweetheart, Kordia, and ‘Lapins’ cherries, the processed fruit had significantly higher procyanidin B2 content than the fresh fruit. For procyanidin B1, processed Kordia (1.28 mg/100 g) contained significantly higher amounts than the other cherry cultivars (Table 4). However, fresh Staccato was higher in procyanidin B1 concentration than fresh Kordia. Additionally, procyanidin B5 and procyanidin B7 were in lower concentrations, ranging from 0.07 to 0.57 mg/100 g and 0.06 to 0.44 mg/100 g, respectively.

Anthocyanins are water-soluble pigments responsible for the characteristic red color of these stone fruits [28,53,54]. In addition to cyanidin glycosides, peonidin derivatives (peonidin glucoside and rutinoside) have also been reported in some sweet cherry cultivars [55]. Cyanidin-3-rutinoside was the major anthocyanin found in this study. Processed Sweetheart contained significantly higher (*p* < 0.05) cyanidin-3-rutinoside content, followed by other fresh and processed cherry cultivars. Processed Kordia and Staccato cherries also exhibited significantly higher (*p* < 0.05) cyanidin-3-rutinoside content than their fresh counterparts (*p* < 0.05). Fresh Staccato, Sweetheart, and ‘Lapins’ cultivars had significantly higher (*p* < 0.05) cyanidin-3-glucoside content than their processed counterparts, whereas processed Kordia cherries exhibited relatively higher (*p* < 0.05) cyanidin-3-glucoside content than fresh Kordia. Processed Sweetheart cherries displayed a significant (*p* < 0.05) amount of peonidin-3-rutinoside, surpassing those of other cherry cultivars. These results support previous findings that the dominant anthocyanins, including cyanidin-3-rutinoside, cyanidin-3-glucoside, and peonidin-3-rutinoside, can vary depending on factors such as ripening stage, genotype, year of study, agronomic practices, and storage conditions [47,51]. In the current study, the analyzed anthocyanins were present in varying concentrations, with cyanidin-3-rutinoside ranging from 1.26 mg/100 g (‘Rainier’) to 80.42 mg/100 g (Sweetheart), cyanidin-3-glucoside ranging from not detected (nd) (‘Rainier’) to 8.81 mg/100 g (Kordia), and peonidin-3-rutinoside ranging from nd (‘Rainier’) to 10.13 mg/100 g (Sweetheart), consistent with findings reported previously [3,28,47,51].

## 4. Conclusions

This study provides a comprehensive analysis of the nutritional and bioactive profile of six sweet cherry cultivars—Kordia, ‘Lapins’, Sweetheart, Staccato, ‘Bing’, and ‘Rainier’, cultivated in New Zealand—in both fresh and processed forms. The findings show that all six cultivars contain various amounts of minerals, phenolic metabolites, and other essential nutrients, with minor variations between fresh and processed forms. Notably, fresh cherries exhibited slightly higher potassium, calcium, and phosphorus concentrations, while processed cherries consistently had more major phenolic metabolites, such as 3-p-coumaroylquinic acid and neochlorogenic acid. Processed ‘Lapins’ and ‘Bing’ cherries showed exceptionally high concentrations of neochlorogenic acid, indicating that processing does not adversely affect this bioactive component. These differences likely reflect specific stages in the processing workflow. For example, water-soluble minerals such as potassium may be partially lost during de-pitting and handling, whereas freeze-drying and matrix disruption can concentrate or enhance the extractability of phenolic metabolites. Packaging and rapid freezing immediately after harvest also help preserve bioactive compounds, suggesting that both the type and sequence of processing steps play a key role in determining the final nutritional and functional composition of the cherries. Overall, this research provides valuable insights into the nutritional and bioactive composition of cherries grown in New Zealand. The findings suggest that cherries, in various forms, can contribute meaningfully to a balanced diet and healthy lifestyle. Additionally, the study highlights opportunities for further investigation into processing methods that may preserve or enhance these beneficial compounds. While the results are specific to the Otago region, future research could explore whether similar nutritional profiles are found in cherries grown in other parts of New Zealand or other parts of the world.

While this study offers valuable insights into the nutritional and bioactive composition of New Zealand–grown cherries, several limitations should be considered. The analyses were performed on a restricted number of cultivars obtained from a single orchard during one growing season, which may limit the extrapolation of findings to other regions, seasons, or climatic conditions. Despite the use of standardized processing and analytical protocols, subtle variations in freeze-drying performance, storage duration, and packaging conditions could have introduced minor experimental variability. Moreover, environmental and agronomic parameters, such as soil composition, fertilization regime, and irrigation practices, were not comprehensively characterized. Finally, functional and in vivo evaluations of the observed bioactivities were beyond the present scope. Future studies should aim to incorporate multi-site sampling, extended cultivar coverage, and biological validation to strengthen the understanding of the nutritional and health-promoting potential of New Zealand cherries.

## Figures and Tables

**Figure 1 foods-14-03749-f001:**
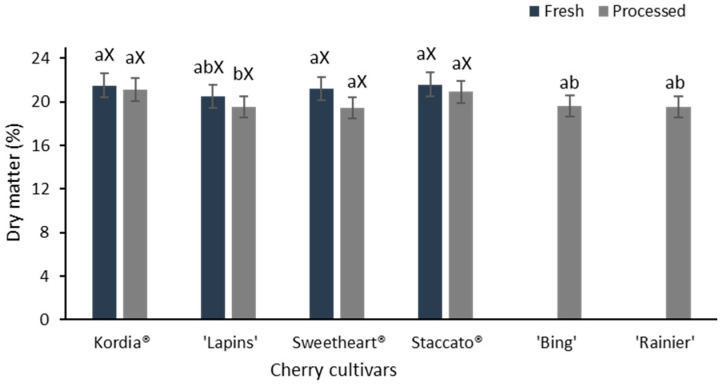
Dry matter content of the six cherry cultivars analyzed in this experiment. (Note: fresh samples were not provided for ‘Bing’ and ‘Rainier’ cultivars). Superscript lowercase letters (a, b) indicate significant differences among all cultivars (n = 6, Tukey’s test, *p* < 0.05), whereas the superscript uppercase letter (X) indicates no significant variation (*p* > 0.05, n = 4) between the fresh and processed samples of the same cultivar.

**Figure 2 foods-14-03749-f002:**
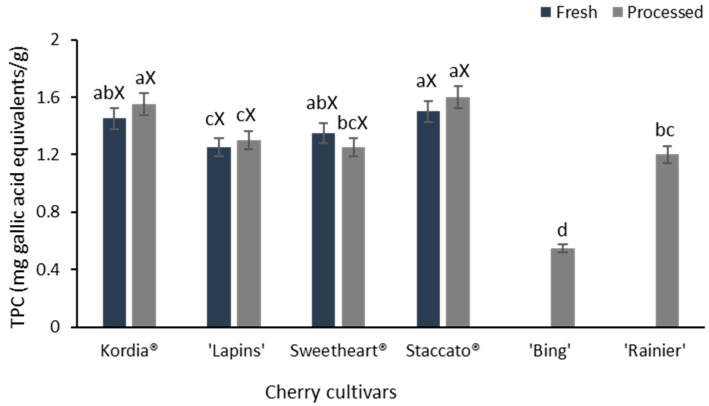
Total phenolic content TPC (mg gallic acid equivalents/g) of fresh and processed cherry samples on the dry weight basis (Note: fresh samples were not available for ‘Bing’ and ‘Rainier’ cultivars). Superscript lowercase letters (a–d) indicate significant differences among all cultivars (n = 6, Tukey’s test, *p* < 0.05), whereas the superscript uppercase letter (X) indicates no significant variation (*p* > 0.05, n = 4) between the fresh and processed samples of the same cultivar.

**Table 1 foods-14-03749-t001:** Nutritional composition of different cultivars of fresh and processed cherry samples on a fresh weight basis.

Nutrient	Kordia^®^		‘Lapins’		Sweetheart^®^		Staccato^®^		‘Bing’	‘Rainier’
	Fresh	Processed	Fresh	Processed	Fresh	Processed	Fresh	Processed	Processed	Processed
**Ash (g/100 g)**	0.5 ± 0.01 ^aX^	0.3 ± 0.02 ^cY^	0.3 ± 0.02 ^cX^	0.3 ± 0.02 ^cY^	0.3 ± 0.02 ^cX^	0.3 ± 0.02 ^cX^	0.4 ± 0.02 ^bX^	0.4 ± 0.02 ^bX^	0.2 ± 0.01 ^d^	0.3 ± 0.02 ^c^
**Fat (g/100 g)**	0.6 ± 0.01 ^cY^	0.8 ± 0.04 ^aX^	0.5 ± 0.01 ^dX^	0.6 ± 0.03 ^cX^	0.6 ± 0.03 ^cX^	0.6 ± 0.03 ^cX^	0.6 ± 0.03 ^cX^	0.5 ± 0.03 ^dX^	0.5 ± 0.03 ^d^	0.7 ± 0.04 ^b^
**Protein (g/100 g)**	1.0 ± 0.02 ^aX^	0.8 ± 0.04 ^cX^	0.7 ± 0.01 ^dX^	0.8 ± 0.05 ^bX^	1.1 ± 0.06 ^aX^	0.8 ± 0.04 ^cX^	0.8 ± 0.04 ^cX^	0.8 ± 0.04 ^bX^	0.9 ± 0.05 ^b^	0.9 ± 0.05 ^b^
**Total available Carb (g/100 g)**	17 ± 0.34 ^bX^	16.7 ± 0.84 ^bX^	16.8 ± 0.34 ^cX^	15.2 ± 0.76 ^cX^	17.7 ± 0.89 ^bX^	17.5 ± 0.88 ^bX^	18.6 ± 0.93 ^aX^	17.3 ± 0.87 ^aX^	16.3 ± 0.82 ^c^	15.1 ± 0.76 ^d^
**TDF (g/100 g)**	1.2 ± 0.02 ^cX^	1.3 ± 0.06 ^aX^	0.8 ± 0.01 ^eX^	1.1 ± 0.06 ^bX^	1.1 ± 0.05 ^dX^	1.1 ± 0.06 ^bX^	1.0 ± 0.06 ^dX^	1.1 ± 0.06 ^bX^	1.1 ± 0.06 ^b^	1.2 ± 0.06 ^a^
**SDF (g/100 g)**	0.4 ± 0.01 ^aX^	0.6 ± 0.03 ^aX^	0.2 ± 0.0 ^cX^	0.5 ± 0.03 ^aX^	0.5 ± 0.03 ^bX^	0.5 ± 0.03 ^aX^	0.3± 0.02 ^aX^	0.5 ± 0.03 ^aX^	0.5 ± 0.03 ^a^	0.6 ± 0.03 ^a^
**IDF (g/100 g)**	0.8 ± 0.02 ^aX^	0.7 ± 0.04 ^bX^	0.6 ± 0.01 ^cX^	0.6 ± 0.03 ^cX^	0.6 ± 0.03 ^cX^	0.6 ± 0.03 ^cX^	0.7 ± 0.04 ^bX^	0.6 ± 0.03 ^cX^	0.6 ± 0.03 ^c^	0.6 ± 0.03 ^c^
**Vit C (mg/100 g)**	4.42 ± 0.22 ^bX^	4.64 ± 0.23 ^bX^	4.13 ± 0.21 ^bX^	4.09 ± 0.2 ^bX^	4.92 ± 0.25 ^bX^	5.32 ± 0.27 ^aX^	5.7 ± 0.29 ^aX^	5.5 ± 0.28 ^aX^	3.13 ± 0.16 ^c^	3.83 ± 0.19 ^c^
**Vit B1 (mg/100 g)**	0.03 ± 00 ^aX^	0.02 ± 00 ^bX^	0.02 ± 00 ^bX^	0.02 ± 00 ^bX^	0.03 ± 00 ^aX^	0.02 ± 00 ^bX^	0.03 ± 00 ^aX^	0.02 ± 0 ^bX^	0.02 ± 00 ^b^	0.03 ± 00 ^a^
**Vit B2 (mg/100 g)**	0.04 ± 00 ^aX^	0.04 ± 00 ^aX^	0.03 ± 00 ^bX^	0.03 ± 00 ^bX^	0.04 ± 00 ^aX^	0.04 ± 00 ^aX^	0.04 ± 00 ^aX^	0.03 ±0 ^bX^	0.03 ± 00 ^b^	0.03 ± 00 ^b^
**Vit B3 (mg/100 g)**	0.05 ± 00 ^d^	<0.01	0.06 ±00 ^cX^	0.05 ± 00 ^dX^	0.07 ± 00 ^b^	<0.01	0.06 ± 00 ^c^	<0.01	0.15 ± 0.01 ^a^	0.03 ± 00 ^e^
**Vit B6 (mg/100 g)**	0.03 ± 00 ^eX^	0.03 ± 0 ^eX^	0.07 ± 00 ^cX^	0.07 ± 00 ^cX^	0.09 ± 00 ^aX^	0.07 ± 00 ^cX^	0.08 ± 00 ^bX^	0.08 ± 0 ^aX^	0.07 ± 00 ^c^	0.06 ± 00 ^d^
**Vit E, ATE (mg/100 g)**	0.2 ± 0.01 ^aX^	0.2 ± 0.01 ^aX^	0.1 ± 0.01 ^bX^	0.1 ± 0.01 ^bX^	0.1 ± 0.01 ^bX^	0.2 ± 0.01 ^aX^	0.2 ± 0.01 ^aX^	0.2 ± 0.01 ^aX^	0.1 ± 0.01 ^b^	0.1 ± 0.01 ^b^
**Vit K (mg/100 g)**	<0.003	0.00391 ± 0.0002 ^a^	<0.003	<0.003	<0.003	<0.003	<0.003	<0.003	<0.003	<0.003
**Vit A, RE (mg/100 g)**	<0.0025	<0.0025	0.0038 ± 0.00019 ^b^	<0.0025	0.0055 ± 0.00028 ^a^	<0.0025	<0.0025	<0.0025	ND	<0.0025
**Calcium (mg/100 g)**	11.6 ± 5.8 ^iX^	12.2 ± 6.2 ^hX^	11.3 ± 5.65 ^jY^	13.1 ± 6.5 ^cX^	12.5 ± 6.25 ^gX^	13.3 ± 6.5 ^dX^	14.8 ± 7.4 ^bX^	14.2 ± 7.1 ^aX^	12.6 ± 6.3 ^f^	12.8 ± 6.4 ^e^
**Magnesium (mg/100 g)**	11.2 ± 5.6 ^cX^	9.9 ± 4.9 ^gY^	9.5 ± 4.75 ^hY^	10.3 ± 5.2 ^dX^	12.0 ± 6.0 ^aX^	10.7 ± 5.4 ^dY^	10.5 ± 5.25 ^fY^	11.0 ± 5.5 ^bX^	10.6 ± 5.3 ^e^	11.2 ± 5.6 ^c^
**Potassium (mg/100 g)**	220.0 ± 110 ^aX^	192.0 ± 96 ^cY^	184.0 ± 92 ^fY^	188.0 ± 94 ^eX^	220.0 ± 110 ^aX^	190.0 ± 95 ^dY^	220.0 ± 110 ^aX^	220.0 ± 110 ^aX^	177.0 ± 88.5 ^g^	200.0 ± 100 ^b^
**Phosphorus (mg/100 g)**	24.0 ± 12 ^bX^	21.0 ± 10.5 ^eY^	19.8 ± 9.9 ^fX^	18.6 ± 9.3 ^fX^	25.0 ± 12.5 ^aX^	21.0 ± 10.5 ^dY^	21.0 ± 10.5 ^eY^	22.0 ± 11 ^cX^	18.3 ± 9.15 ^g^	21.0 ± 10.5 ^e^
**Iron (mg/100 g)**	0.26 ± 0.13 ^bX^	0.19 ± 0.01 ^fY^	0.18 ± 0.09 ^hY^	0.20 ± 0.10 ^eX^	0.28 ± 0.14 ^aX^	0.21 ± 0.11 ^dY^	0.18 ± 0.09 ^hY^	0.20 ± 0.10 ^eX^	0.23 ± 0.12 ^c^	0.19 ± 0.10 ^g^
**Copper (mg/100 g)**	0.13 ± 0.06 ^cX^	0.104 ± 0.05 ^eY^	0.195 ± 0.01 ^aX^	0.083 ± 0.04 ^eY^	0.137 ± 0.07 ^bX^	0.083 ± 0.04 ^eY^	0.083 ± 0.04 ^fY^	0.105 ± 0.05 ^dX^	0.082 ± 0.04 ^g^	0.081 ± 0.04 ^h^
**Manganese (mg/100 g)**	0.066 ± 0.03 ^eX^	0.06 ± 0.03 ^gY^	0.051 ± 0.03 ^hY^	0.058 ± 0.03 ^gX^	0.065 ± 0.03 ^fY^	0.104 ± 0.05 ^bX^	0.072 ± 0.04 ^dY^	0.092 ± 0.05 ^cX^	0.119 ± 0.06 ^a^	0.062 ± 0.03 ^g^
**Zinc (mg/100 g)**	0.07 ± 0.04 ^cX^	0.06 ± 0.03 ^cX^	0.13 ± 0.07 ^aX^	0.09 ± 0.05 ^bY^	0.13 ± 0.07 ^aX^	0.13 ± 0.07 ^aX^	0.09 ± 0.05 ^bY^	0.09 ± 0.05 ^bY^	0.09 ± 0.05 ^b^	0.08 ± 0.04 ^c^

Note: Carb = carbohydrates; Vit = vitamin; IDF = insoluble dietary fiber; SDF = soluble dietary fiber; TDF = total dietary fiber. ‘Bing’ and ‘Rainier’ fresh samples were not available. Hence, only the results for processed samples are presented. Superscript lowercase letters (a–j) indicate significant differences (*p* < 0.05, n = 6) across the samples of all cultivars, whereas the superscript uppercase letters (X and Y) indicate the significant differences (*p* < 0.05, Tukey’s test, n = 4) between the processed and fresh samples of the same cultivar. ATE = alpha-tocopherol equivalents, RE = retinol equivalents.

**Table 2 foods-14-03749-t002:** Sugar composition of fresh and processed cherry samples on the fresh weight basis.

Sugar (g/100 g)	Kordia^®^		‘Lapins’		Sweetheart^®^		Staccato^®^		‘Bing’	‘Rainier’
	Fresh	Processed	Fresh	Processed	Fresh	Processed	Fresh	Processed	Processed	Processed
**Glucose**	7.3 ± 0.15 ^bX^	7.3 ± 0.15 ^bX^	7.1 ± 0.14 ^bX^	6.6 ± 0.13 ^cX^	7.9 ± 0.16 ^aX^	7.5 ± 0.15 ^bX^	8.0 ± 0.16 ^aX^	7.4 ± 0.15 ^bY^	6.8 ± 0.14 ^c^	6.6 ± 0.13 ^c^
**Fructose**	6.5 ± 0.13 ^aX^	6.5 ± 0.13 ^aX^	6.2 ±0.12 ^aX^	5.7 ± 0.11 ^bX^	6.6 ± 0.13 ^aX^	6.0 ± 0.12 ^bY^	6.6 ± 0.13 ^aX^	6.2 ± 0.12 ^aX^	6.1 ± 0.12 ^b^	5.8 ± 0.12 ^c^
**Maltose**	<0.1	<0.1	<0.1	<0.1	<0.1	<0.1	<0.1	<0.1	<0.1	<0.1
**Sucrose**	<0.1	<0.1	<0.1	<0.1	<0.1	<0.1	<0.1	<0.1	<0.1	<0.1
**Galactose**	<0.1	<0.1	<0.1	<0.1	<0.1	<0.1	<0.1	<0.1	<0.1	<0.1
**Total sugar**	13.8 ± 0.28 ^aX^	13.8 ± 0.28 ^aX^	13.2 ± 0.26 ^bX^	12.2 ± 0.24 ^bX^	14.5 ± 0.29 ^aX^	13.1 ± 0.26 ^bY^	14.5 ± 0.29 ^aX^	13.6 ± 0.27 ^aX^	12.9 ± 0.26 ^c^	12.4 ± 0.25 ^c^

Note: ‘Bing’ and ‘Rainier’ fresh samples were not available; hence, only the results for processed samples are presented. Superscript lowercase letters (a–c indicate significant differences (*p* < 0.05, n = 6) across the samples of all cultivars, whereas the superscript uppercase letters (X and Y) indicate the significant differences (*p* < 0.05, Tukey’s test, n = 4) between the processed and fresh samples of the same cultivar.

**Table 3 foods-14-03749-t003:** Amino acid composition of fresh and processed cherry samples on the fresh weight basis.

Amino Acid (mg/g)	Kordia^®^		‘Lapins’		Sweetheart^®^		Staccato^®^		Bing’	‘Rainier’
	Fresh	Processed	Fresh	Processed	Fresh	Processed	Fresh	Processed	Processed	Processed
**Aspartic Acid**	3.75 ± 0.19 ^aX^	2.19 ± 0.11 ^cY^	1.52 ± 0.08 ^eY^	2.11 ± 0.11 ^cX^	3.61 ± 0.18 ^aX^	1.74 ± 0.09 ^dY^	1.59 ± 0.08 ^bY^	2.0 ± 0.1 ^cX^	3.01 ± 0.15 ^b^	2.79 ± 0.14 ^b^
**Threonine**	0.24 ± 0.01 ^aX^	0.22 ± 0.01 ^aY^	0.18 ± 0.01 ^cY^	0.21 ± 0.01 ^aX^	0.24 ± 0.01 ^aX^	0.22 ± 0.01 ^aX^	0.23 ± 0.01 ^aX^	0.21 ± 0.01 ^aX^	0.19 ± 0.01 ^b^	0.2 ± 0.01 ^b^
**Serine**	0.28 ± 0.01 ^aX^	0.25 ± 0.01 ^aX^	0.2 ± 0.01 ^cY^	0.25 ± 0.01 ^aX^	0.26 ± 0.01 ^aX^	0.25 ± 0.01 ^aX^	0.28 ± 0.01 ^aX^	0.24 ± 0.01 ^aX^	0.22 ± 0.01 ^b^	0.21 ± 0.01 ^b^
**Glutamic Acid**	0.65 ± 0.03 ^aX^	0.56 ± 0.03 ^bY^	0.44 ± 0.02 ^dY^	0.52 ± 0.03 ^bX^	0.62 ± 0.03 ^aX^	0.54 ± 0.03 ^bY^	0.53 ± 0.03 ^bX^	0.51 ± 0.03 ^bX^	0.59 ± 0.03 ^a^	0.52 ± 0.03 ^c^
**Proline**	0.49 ± 0.02 ^bX^	0.40 ± 0.02 ^dY^	0.33 ± 0.02 ^eX^	0.33 ± 0.02 ^eX^	0.57 ± 0.03 ^aX^	0.39 ± 0.02 ^dY^	0.45 ± 0.02 ^cX^	0.40 ± 0.02 ^cX^	0.44 ± 0.02 ^c^	0.35 ± 0.02 ^e^
**Glycine**	0.26 ± 0.01 ^aX^	0.25 ± 0.01 ^aX^	0.23 ± 0.01 ^bY^	0.24 ± 0.01 ^aX^	0.25 ± 0.01 ^aX^	0.25 ± 0.01 ^aX^	0.26 ± 0.01 ^aX^	0.24 ± 0.01 ^aX^	0.22 ± 0.01 ^b^	0.21 ± 0.01 ^b^
**Alanine**	0.24 ± 0.01 ^aX^	0.24 ± 0.01 ^aX^	0.2 ± 0.01 ^bY^	0.23 ± 0.01 ^aX^	0.23 ± 0.01 ^bY^	0.24 ± 0.01 ^aX^	0.24 ± 0.01 ^aX^	0.23 ± 0.01 ^aX^	0.21 ± 0.01 ^b^	0.21 ± 0.01 ^b^
**Valine**	0.23 ± 0.01 ^aX^	0.23 ± 0.01 ^aX^	0.2 ± 0.01 ^bY^	0.22 ± 0.01 ^aX^	0.23 ± 0.01 ^aX^	0.23 ± 0.01 ^aX^	0.23 ± 0.01 ^aX^	0.22 ± 0.01 ^aX^	0.2 ± 0.01 ^b^	0.2 ± 0.01 ^b^
**Isoleucine**	0.33 ± 0.02 ^aX^	0.3 ± 0.01 ^aX^	0.23 ± 0.01 ^cY^	0.29 ± 0.01 ^aX^	0.3 ± 0.02 ^aX^	0.26 ± 0.01 ^bY^	0.31 ± 0.02 ^aX^	0.25 ± 0.01 ^bY^	0.25 ± 0.01 ^b^	0.26 ± 0.01 ^b^
**Leucine**	0.30 ± 0.02 ^aX^	0.31 ± 0.02 ^aX^	0.27 ± 0.01 ^aX^	0.28 ± 0.01 ^aX^	0.29 ± 0.01 ^aX^	0.30 ± 0.02 ^aX^	0.30 ± 0.02 ^aX^	0.29 ± 0.01 ^aX^	0.26 ± 0.01 ^b^	0.26 ± 0.01 ^b^
**Tyrosine**	0.19 ± 0.01 ^aX^	0.18 ± 0.01 ^aX^	0.16 ± 0.01 ^aY^	0.18 ± 0.01 ^aX^	0.19 ± 0.01 ^aX^	0.18 ± 0.01 ^aX^	0.19 ± 0.01 ^aX^	0.18 ± 0.01 ^aX^	0.16 ± 0.01 ^Y^	0.16 ± 0.01 ^Y^
**Phenylalanine**	0.21 ± 0.01 ^aX^	0.21 ± 0.01 ^aX^	0.18 ± 0.01 ^bY^	0.21 ± 0.01 ^aX^	0.21 ± 0.01 ^aX^	0.21 ± 0.01 ^aX^	0.22 ± 0.01 ^aX^	0.21 ± 0.01 ^aX^	0.18 ± 0.01 ^bY^	0.18 ± 0.01 ^bY^
**Histidine**	0.13 ± 0.01 ^aX^	0.12 ± 0.01 ^aX^	0.09 ± 0.01 ^bY^	0.10 ± 0.01 ^aX^	0.13 ± 0.01 ^aX^	0.11 ± 0.01 ^aX^	0.11 ± 0.01 ^aX^	0.11 ± 0.01 ^aX^	0.10 ± 0.01 ^a^	0.10 ± 0.01 ^a^
**Lysine**	0.30 ± 0.02 ^aX^	0.31 ± 0.02 ^aX^	0.27 ± 0.01 ^bY^	0.28 ± 0.02 ^aX^	0.31 ± 0.02 ^aX^	0.31 ± 0.02 ^aX^	0.30 ± 0.02 ^aX^	0.30 ± 0.02 ^aX^	0.26 ± 0.01 ^b^	0.26 ± 0.01 ^b^
**Arginine**	0.21 ± 0.01 ^aX^	0.20 ± 0.01 ^aX^	0.17 ± 0.01 ^bX^	0.19 ± 0.01 ^aX^	0.20 ± 0.01 ^aX^	0.19 ± 0.01 ^aX^	0.20 ± 0.01 ^aX^	0.18 ± 0.01 ^aX^	0.18 ± 0.01 ^a^	0.17 ± 0.01 ^a^
**Cysteine**	0.08 ± 0.01 ^bY^	0.07 ± 0.01 ^bY^	0.07 ± 0.01 ^bY^	0.07 ± 0.01 ^bY^	0.11 ± 0.01 ^aX^	0.10 ± 0.01 ^aX^	0.10 ± 0.01 ^aX^	0.09 ± 0.01 ^aX^	0.05 ± 0.01 ^c^	0.06 ± 0.01 ^c^
**Methionine**	0.12 ± 0.01 ^cX^	0.11 ± 0.01 ^cX^	0.11 ± 0.01 ^cX^	0.13 ± 0.01 ^aX^	0.13 ± 0.01 ^bX^	0.18 ± 0.01 ^aX^	0.15 ± 0.01 ^aX^	0.15 ± 0.01 ^aX^	0.12 ± 0.01 ^c^	0.12 ± 0.01 ^c^
**Tryptophan**	0.06 ± 0.01 ^bY^	0.04 ± 0.01 ^bY^	0.04 ± 0.01 ^bY^	0.05 ± 0.01 ^bY^	0.09 ± 0.01 ^aX^	0.05 ± 0.01 ^bY^	0.05 ± 0.01 ^bY^	0.09 ± 0.01 ^aX^	0.05 ± 0.01 ^b^	0.05 ± 0.01 ^b^

Note: ‘Bing’ and ‘Rainier’ fresh samples were not available. Hence, only the results for processed samples are presented. Superscript lowercase letters (a–e) indicate significant differences (*p* < 0.05, n = 6) across the samples of all cultivars, whereas the superscript uppercase letters (X and Y) indicate the significant differences (*p* < 0.05, Tukey’s test, n = 4) between the processed and fresh samples of the same cultivar.

**Table 4 foods-14-03749-t004:** Composition of phenolic metabolites (grouped by classification) in fresh and processed cherry samples as fresh weight (mg/100 g FW) basis.

Phenolic Metabolite (mg/100 g FW)	Kordia^®^		‘Lapins’		Staccato^®^		Sweetheart^®^		‘Bing’	‘Rainier’
	Fresh	Processed	Fresh	Processed	Fresh	Processed	Fresh	Processed	Processed	Processed
**Hydroxycinnamic acid**										
3,5-Dicaffeoylquinic acid	1.85	1.83	1.99	2.29	2.28	2.43	1.61	2.02	2.70	1.16
3-*p*-Coumaroyl quinic acid	21.41	34.21	3.70	5.35	26.27	22.57	15.93	19.00	6.68	1.95
Chlorogenic acid	2.27	3.68	2.14	3.21	3.15	3.00	1.86	3.02	3.52	1.79
Ferulic acid	nd	nd	0.09	0.11	nd	nd	nd	nd	0.08	0.08
Neochlorogenic acid	20.47	34.51	28.05	40.96	29.50	27.72	17.87	25.10	44.26	22.78
*trans*-4-*p*-Coumaroyl quinic acid	nd	0.63	0.08	0.12	0.99	0.96	0.47	0.85	0.12	0.05
*trans*-5-*p*-Coumaroyl quinic acid	0.13	0.16	0.08	0.14	0.11	0.13	nd	0.08	0.14	0.07
**Flavanol**										
Catechin	1.32	1.76	0.60	0.65	0.84	0.83	0.57	0.81	0.83	0.34
Epicatechin	2.97	4.14	2.51	3.74	7.89	7.25	3.76	4.89	3.94	0.95
**Flavonol**										
Kaempferol 3-rutinoside	0.45	0.80	0.20	0.45	0.61	0.55	0.29	0.57	0.47	0.64
Quercetin 3-galactoside	nd	nd	nd	nd	nd	nd	nd	0.01	nd	nd
Quercetin 3-glucoside	0.55	0.73	0.16	0.22	0.39	0.25	0.30	0.17	0.26	0.06
Quercetin 3-rhamnoside	0.03	0.06	0.01	0.03	0.04	0.03	0.02	0.02	0.03	0.02
Quercetin 3-rutinoside	3.45	4.34	1.99	2.38	3.24	2.68	2.66	2.93	2.46	1.03
**Flavanone**										
Sakuranetin	0.03	0.03	0.00	0.01	0.00	0.00	0.01	0.01	0.00	0.01
Sakuranin	0.32	0.41	0.08	0.12	0.19	0.11	0.09	0.09	0.10	0.08
**Procyanidin**										
Procyanidin B1	0.94	1.28	0.46	0.48	1.04	1.00	0.60	0.91	0.59	0.14
Procyanidin B2	1.14	1.70	1.03	1.56	4.16	3.34	1.76	2.80	1.50	0.37
Procyanidin B5	0.21	0.29	0.16	0.24	0.57	0.44	0.22	0.31	0.22	0.07
Procyanidin B7	0.32	0.43	0.16	0.21	0.44	0.36	0.22	0.24	0.21	0.06
**Anthocyanin**										
Cyanidin 3-glucoside	6.73	8.81	3.14	2.93	8.77	6.19	7.39	3.85	3.97	nd
Cyanidin 3-rutinoside	48.44	58.77	45.90	40.16	68.30	66.33	65.98	80.42	46.55	1.26
Peonidin 3-rutinoside	3.09	3.28	4.28	4.04	6.28	6.48	8.83	10.13	4.29	nd

Note: ‘Bing’ and ‘Rainier’ fresh samples were not available. Hence, only the results for processed samples are presented. nd: not detected.

## Data Availability

The data presented in this study are available on request from the corresponding author due to privacy.

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
