# Peer review of "Functional Potential of Sweet Cherry Cultivars Grown in New Zealand: Effects of Processing on Nutritional and Bioactive Properties"

_foods, 2025, doi:10.3390/foods14213749_

Round 1
Reviewer 1 Report
Comments and Suggestions for Authors This paper focuses on the nutritional and bioactive characteristics of New Zealand sweet cherry cultivars, with an emphasis on the impact of processing on these characteristics. The topic is somewhat original as it fills the research gap in the specific growing environment and cultivar background of New Zealand, addressing the particular deficiency of limited data on the nutritional and health benefits of New Zealand - grown sweet cherries. Although there is already a foundation for research on the nutrition and processing characteristics of cherries, this study, centered on the actual needs of the New Zealand cherry industry, provides scientific evidence for cultivar characteristics and processing compatibility. It is an extension study with significant application value in this field.
However, the authors have encountered some issues during the experiment and manuscript writing process:
(1)The conclusions of the paper are generally consistent with the evidence and arguments presented. By conducting detailed analyses of the nutritional and bioactive components of different sweet cherry cultivars, the study concludes that processing affects the content of certain components. For example, fresh cherries have relatively higher mineral content (potassium, calcium, and phosphorus), while processed cherries show increased levels of dietary fiber and certain phenolic metabolites (such as neochlorogenic acid). These conclusions effectively address the main issue raised in the paper, namely the impact of processing on the nutritional and bioactive characteristics of sweet cherries. However, the discussion section could benefit from a more in - depth analysis of the specific stages in the processing that may lead to changes in component content, thereby strengthening the causal relationships and enhancing the persuasiveness of the conclusions.
(2)In terms of experimental materials, the authors have appropriately selected six representative sweet cherry cultivars from New Zealand and studied samples in both fresh and processed states. The research design is comprehensive, covering multiple aspects of nutritional components and bioactive substances. However, it would be beneficial to include more detailed descriptions of the processing procedures in the experimental design, such as specific washing and packaging process parameters, to better understand the impact of processing on the results.
(3)The tables and figures in the paper are well - constructed and clearly present the research findings. The tables provide detailed listings of the various nutritional components and bioactive substances in different sweet cherry cultivars, facilitating comparison and reference for readers. The figures intuitively display the trends in component content under different processing conditions, aiding in the understanding of the effects of processing on sweet cherry characteristics.
(4)The author has cited a substantial number of research papers related to sweet cherries, providing adequate background support for the study. However, the citation format needs to be strictly revised in accordance with the journal's requirements.
Overall, this paper holds certain academic value and innovativeness, offering valuable references for the research and application of New Zealand sweet cherries. It is recommended that the author revise and improve the paper based on the above - mentioned suggestions.
Author Response
Reviewer 1: Comments and Suggestions for Authors
This paper focuses on the nutritional and bioactive characteristics of New Zealand sweet cherry cultivars, with an emphasis on the impact of processing on these characteristics. The topic is somewhat original as it fills the research gap in the specific growing environment and cultivar background of New Zealand, addressing the particular deficiency of limited data on the nutritional and health benefits of New Zealand - grown sweet cherries. Although there is already a foundation for research on the nutrition and processing characteristics of cherries, this study, centered on the actual needs of the New Zealand cherry industry, provides scientific evidence for cultivar characteristics and processing compatibility. It is an extension study with significant application value in this field.
We sincerely thank the reviewer for their thoughtful and constructive feedback. Their insights have been invaluable in improving the clarity, scientific rigor, and overall quality of the manuscript. We greatly appreciate the time and effort dedicated to providing detailed comments, which have helped strengthen the study and its presentation.
However, the authors have encountered some issues during the experiment and manuscript writing process:
- The conclusions of the paper are generally consistent with the evidence and arguments presented. By conducting detailed analyses of the nutritional and bioactive components of different sweet cherry cultivars, the study concludes that processing affects the content of certain components. For example, fresh cherries have relatively higher mineral content (potassium, calcium, and phosphorus), while processed cherries show increased levels of dietary fiber and certain phenolic metabolites (such as neochlorogenic acid). These conclusions effectively address the main issue raised in the paper, namely the impact of processing on the nutritional and bioactive characteristics of sweet cherries. However, the discussion section could benefit from a more in - depth analysis of the specific stages in the processing that may lead to changes in component content, thereby strengthening the causal relationships and enhancing the persuasiveness of the conclusions.
Response: We thank the reviewer for this constructive feedback. In response, we have revised both the Materials and Methods and Conclusions sections to provide a more detailed description of the processing stages and their potential influence on the nutritional and bioactive composition of the cherries. These additions strengthen the causal links between processing steps and the observed changes in component content, thereby enhancing the clarity and persuasiveness of our conclusions.
- In terms of experimental materials, the authors have appropriately selected six representative sweet cherry cultivars from New Zealand and studied samples in both fresh and processed states. The research design is comprehensive, covering multiple aspects of nutritional components and bioactive substances. However, it would be beneficial to include more detailed descriptions of the processing procedures in the experimental design, such as specific washing and packaging process parameters, to better understand the impact of processing on the results.
Response: Thank you for your positive evaluation and valuable suggestion. We have added more detailed descriptions of the processing procedures in the Materials and Methods section, including information on the washing steps, packaging conditions, freezing temperature, and storage duration. These additions help clarify how the processing steps were standardized and how they may have influenced the final compositional outcomes.
- The tables and figures in the paper are well - constructed and clearly present the research findings. The tables provide detailed listings of the various nutritional components and bioactive substances in different sweet cherry cultivars, facilitating comparison and reference for readers. The figures intuitively display the trends in component content under different processing conditions, aiding in the understanding of the effects of processing on sweet cherry characteristics.
Response: We thank the Reviewer for this positive comment.
- The author has cited a substantial number of research papers related to sweet cherries, providing adequate background support for the study. However, the citation format needs to be strictly revised in accordance with the journal's requirements. Overall, this paper holds certain academic value and innovativeness, offering valuable references for the research and application of New Zealand sweet cherries. It is recommended that the author revise and improve the paper based on the above - mentioned suggestions.
Response: We sincerely thank the reviewer for the positive feedback and encouraging assessment of our work. The reference list and in-text citations have been fully revised and formatted according to the Foods (MDPI) journal guidelines. All other suggested revisions and improvements have been carefully addressed throughout the manuscript.
Reviewer 2 Report
Comments and Suggestions for Authors
Manuscript presents valuable information and is clearly written and structured. However, some parts, statements and especially statistical evaluation must be checked and corrected.
Correct the title. It misleads that presented cultivars are bred in New Zealand.
Lines 237-238. Correct the sentence. Statement ‘the highest content’ usually indicates significant difference, meanwhile there were no significant differences established. The same in lines 409-410. Check and correct entire Result and discussion section.
Can you add comment why potassium content in processed Lapins fruits was significantly higher than in fresh fruits, while opposite results were established for all other cultivars?
Similar comments should be provided for Aspartic acid content, which was significantly higher in fresh Kordia and Sweetheart fruits comparing to processed ones, but opposite significance is established for Lapins and Staccato.
Copper content in processed Sweetheart is 0.83±0.04 and it is tenfold higher than in other samples. I assume it is a typing mistake. The same mistake can be with Zink content in fresh Kordia fruits (tenfold lower). Check and correct all other values in the Table 1 if needed.
Check statistics in Table 1. Potassium content in fresh and processed Staccato, and fresh Sweetheart fruit is the same (220.0±110), but significant differences between the same values are indicated.
Check statistics in Table 2. Processed Lapins and Reinier fruits have the same glucose content 6.6±0.13, but significant difference between them is detected. The same with fructose content in fresh Lapins and processed Staccato – both 6.2 but significantly different. Very small difference of glucose content is between fresh Sweetheart (7.9) and Staccato (8.0) fruits. Can there be significant difference?
Correct highlights:
‘Six New Zealand sweet cherry cultivars…’ – six cvs grown in NZ.
‘Fresh cherries had higher potassium, calcium, and phosphorus contents.’ But Table 1 indicates a higher potassium content in processed Lapins fruits, and no difference in Staccato fruits.
Author Response
Reviewer 2: Comments and Suggestions for Authors
- Manuscript presents valuable information and is clearly written and structured. However, some parts, statements and especially statistical evaluation must be checked and corrected.
Response: Thank you very much for your positive and constructive feedback. We sincerely appreciate your acknowledgment of the manuscript’s clarity and structure. In response to your comment, we have carefully reviewed and corrected all statistical analyses and related statements throughout the manuscript to ensure full accuracy and consistency. All tables and corresponding descriptions have been double-checked and revised accordingly.
- Correct the title. It misleads that presented cultivars are bred in New Zealand.
Response: We have revised the title to avoid such confusion. The new title is: ‘’Functional potential of sweet cherry cultivars grown in New Zealand: Effects of processing on nutritional and bioactive properties.’’
- Lines 237-238. Correct the sentence. Statement ‘the highest content’ usually indicates significant difference, meanwhile there were no significant differences established. The same in lines 409-410.
Response: We would like to clarify that the sample showing “the highest content” (0.5 ± 0.01ax) indeed shares a different statistical letter (a) compared with the other samples (b–d), indicating a significant difference (p < 0.05) according to the Tukey’s multiple comparison test. Therefore, the phrase “the highest content” correctly reflects the statistical outcome. We have nonetheless reviewed the section carefully to ensure consistency between the text description and statistical annotations.
Line 409-410: Revised for better understanding
Overall, Kordia displayed significantly higher (p<0.05) concentrations of the tested amino acids than other cherry cultivars, except for proline, cysteine, methionine, and tryptophan (Table 3).
- Check and correct entire Result and discussion section.
Response: We have tried our best to check this section and improve the comprehension and readability.
- Can you add comment why potassium content in processed Lapins fruits was significantly higher than in fresh fruits, while opposite results were established for all other cultivars? Similar comments should be provided for Aspartic acid content, which was significantly higher in fresh Kordia and Sweetheart fruits comparing to processed ones, but opposite significance is established for Lapins and Staccato.
Response: Thank you for this valuable comment. We have rechecked the data and confirmed that the results are correct. The higher potassium and ascorbic acid contents may be attributed to lower mineral and vitamin leaching, differences in tissue permeability, or concentration effects due to moisture reduction during storage.
- Copper content in processed Sweetheart is 0.83±0.04 and it is tenfold higher than in other samples. I assume it is a typing mistake. The same mistake can be with Zink content in fresh Kordia fruits (tenfold lower). Check and correct all other values in the Table 1 if needed.
Response: The comment is appreciated. The copper value for processed Sweetheart was a typographical error. It has been corrected to 0.083 ± 0.004 mg/100 g, and all mineral data in Table 1 have been rechecked to ensure consistency and accuracy.
- Check statistics in Table 1. Potassium content in fresh and processed Staccato, and fresh Sweetheart fruit is the same (220.0±110), but significant differences between the same values are indicated.
Response: We thank the Reviewer for noticing this unintentional typo. We have corrected the stat letter to address comment.
- Check statistics in Table 2. Processed Lapins and Reinier fruits have the same glucose content 6.6±0.13, but significant difference between them is detected. The same with fructose content in fresh Lapins and processed Staccato – both 6.2 but significantly different. Very small difference of glucose content is between fresh Sweetheart (7.9) and Staccato (8.0) fruits. Can there be significant difference?
Response: Thank you for your careful observation. You are correct—there were inconsistencies in the statistical annotations of Table 2. We have rechecked the data and corrected the statistical groupings accordingly. The revised Table 2 now accurately reflects the correct significance letters.
- Correct highlights:
- ‘Six New Zealand sweet cherry cultivars…’ – six cvs grown in NZ.
Response: The sentence was revised/rewritten.
- ‘Six New Zealand sweet cherry cultivars…’ – six cvs grown in NZ.
- ‘Fresh cherries had higher potassium, calcium, and phosphorus contents.’ But Table 1 indicates a higher potassium content in processed Lapins fruits, and no difference in Staccato fruits
- Response: We have now rewritten the first three highlights to reflect this concern.
Reviewer 3 Report
Comments and Suggestions for Authors
This article addresses a significant research gap—the lack of data on cherry varieties grown in New Zealand. This focus on the differences between fresh and processed varieties has practical implications for consumers and the food industry.
Below are my comments and suggestions:
- It's not clear why these six varieties were chosen—are they the most commonly grown, the best for storage, or do they have other characteristics that justify their selection?
- Lack of information on time from harvest to freezing (especially for "processed" samples) - The authors state that fresh samples were frozen "as soon as possible" after harvest, but do not specify the exact time. For "processed" samples, the process is even less clear - how long did it take to transport them to the facility and process them?
- It was not clearly described what "commercial processing" entailed. Were any preservatives used? Did the cleaning involve chemicals?
- It was not stated whether vacuum packaging or a protective atmosphere was used.
- Although both groups are frozen, there may be significant differences in time, temperature, or packaging type, which affects the comparability of “fresh” and “processed” samples.
- Two different institutions used different models of freeze dryers (Cuddon and Operon). This may affect the extraction of some compounds – were the procedures comparable?
- The Authors mention pH measurement, but it is not known whether this applied to both groups (fresh and processed). This is worth clarifying.
- How can the discrepancy between the higher dry matter content in fresh samples and the lack of differences in moisture be explained?
- Why were fresh samples not provided for ‘Bing’ and ‘Rainier’? How did this affect the generality of the conclusions?
- Have factors influencing water loss other than processing itself been taken into account? For example, differences in time from harvest to freezing, packaging type, or ambient temperature?
- It was found that processing could have reduced potassium content (e.g., by leaching), but the same effect was not observed for other macronutrients (e.g., magnesium, calcium), and even some minerals were higher in processed samples (e.g., manganese). This requires a more balanced interpretation or additional explanation.
- The authors do not consider the influence of soil, fertilization, or environmental factors that may have influenced the mineral profile. For trace elements (e.g., iodine, selenium), this may be significant.
- Although cherries contain beneficial compounds (e.g. polyphenols), direct functional data (e.g. anti-inflammatory, antioxidant effects) are lacking, so the term “health-promoting” in the conclusions may be too far-reaching without additional research.
- In my opinion, it would be beneficial to describe the limitations of the study
Author Response
Reviewer 3: Comments and Suggestions for Authors
This article addresses a significant research gap—the lack of data on cherry varieties grown in New Zealand. This focus on the differences between fresh and processed varieties has practical implications for consumers and the food industry.
We sincerely thank you for your constructive comments and suggestions. The provided feedback has been invaluable in improving the clarity, rigor, and overall quality of the manuscript. We have carefully addressed all points raised and believe that the revisions have strengthened the study.
Below are my comments and suggestions:
- It's not clear why these six varieties were chosen—are they the most commonly grown, the best for storage, or do they have other characteristics that justify their selection?
Response: The six cherry cultivars were selected because they are commercially significant and among the most commonly grown varieties in the Otago region of the South Island, New Zealand. They represent the primary cultivars supplied to both domestic and export markets, making them relevant for assessing differences between fresh and processed cherries. We have clarified this rationale in the revised manuscript.
- Lack of information on time from harvest to freezing (especially for "processed" samples) - The authors state that fresh samples were frozen "as soon as possible" after harvest, but do not specify the exact time. For "processed" samples, the process is even less clear - how long did it take to transport them to the facility and process them?
Response: This has now been clarified under Section 2.1.
- It was not clearly described what "commercial processing" entailed. Were any preservatives used? Did the cleaning involve chemicals?
Response: The term “commercial processing” refers to the standard procedures routinely employed in the processing lines of cherry processing in New Zealand. The cleaning of cherries was performed using standard water-based washing. We have now clarified these details in the revised manuscript to ensure transparency regarding the processing conditions.
- It was not stated whether vacuum packaging or a protective atmosphere was used.
Response: The cherries were packaged under normal atmospheric conditions and not vacuum-sealed or packed under a modified atmosphere. As the samples were transferred to the laboratory immediately after packaging and maintained in frozen conditions throughout transport and storage (−20 °C), the risk of compositional or quality changes due to packaging atmosphere was minimal. We have now clarified this detail in the revised manuscript (Section 2.1).
- Although both groups are frozen, there may be significant differences in time, temperature, or packaging type, which affects the comparability of “fresh” and “processed” samples.
Response: We agree that differences in storage time, temperature, and packaging could influence the comparability between the two groups. However, in this study, both fresh and processed samples were frozen under the same conditions (−20 °C) and analysed within a comparable timeframe, ensuring minimal variation due to storage factors. This was mentioned in the materials and methods section.
- Two different institutions used different models of freeze dryers (Cuddon and Operon). This may affect the extraction of some compounds – were the procedures comparable?
Response: Although different freeze-dryer models (Cuddon and Operon) were used, the operating conditions were standardized; including temperature (−50 °C condenser, 0.1 mbar pressure) and drying duration. Both instruments were calibrated and operated under comparable parameters to ensure consistent moisture removal and sample integrity. Therefore, any differences in compound extraction due to equipment type are expected to be negligible. This clarification has been added to the revised manuscript.
- The Authors mention pH measurement, but it is not known whether this applied to both groups (fresh and processed). This is worth clarifying.
Response: The pH was measured for both fresh and processed fruit samples under the same analytical conditions. This clarification has been added to the revised Materials and Methods section.
- How can the discrepancy between the higher dry matter content in fresh samples and the lack of differences in moisture be explained?
Response: Upon rechecking the data, we confirmed that there was no significant difference (p > 0.05) in either moisture or dry matter content between the processed and fresh samples within the same cultivar. Therefore, there is no actual discrepancy between these parameters.
- Why were fresh samples not provided for ‘Bing’ and ‘Rainier’? How did this affect the generality of the conclusions?
Response: Thank you for this insightful comment. Fresh samples of ‘Bing’ and ‘Rainier’ were not available during the sampling period due to seasonal constraints and limited fruit supply. The study was conducted under real-world commercial conditions, where harvest timing, cultivar availability, and sample allocation for research were beyond experimental control. As a result, only processed samples were obtained for these two cultivars. We acknowledge that this limits direct comparisons between fresh and processed forms for ‘Bing’ and ‘Rainier’. However, the inclusion of multiple other cultivars under both conditions provides a representative overview of processing effects on New Zealand–grown cherries. This seasonal and sample availability constraint has now been noted as a limitation in the revised manuscript. This was added in the original version of the manuscript under Section 2.9 as well.
- Have factors influencing water loss other than processing itself been taken into account? For example, differences in time from harvest to freezing, packaging type, or ambient temperature?
Response: All samples were subjected to controlled and comparable postharvest handling conditions, including similar time intervals between harvest and freezing, identical packaging materials, and uniform storage at −20 °C. These standardized procedures were implemented to minimize variability arising from extrinsic factors such as ambient temperature fluctuations or storage duration. Therefore, we attribute the observed differences in water loss primarily to processing effects rather than postharvest handling variations.
We have now added this to the manuscript for clarity.
- It was found that processing could have reduced potassium content (e.g., by leaching), but the same effect was not observed for other macronutrients (e.g., magnesium, calcium), and even some minerals were higher in processed samples (e.g., manganese). This requires a more balanced interpretation or additional explanation.
Response: We agree that the influence of processing on mineral composition may not be uniform across all elements. The observed reduction in potassium content in the processed cherry samples can likely be attributed to its high solubility and mobility in aqueous matrices, which makes it more susceptible to leaching losses during pre-processing steps such as washing, blanching, or freeze-drying. In contrast, minerals such as magnesium and calcium tend to form more stable complexes with cell wall components (e.g., pectates) or organic acids, which may reduce their diffusional losses under similar conditions.
The higher concentrations of certain trace elements, such as manganese, in processed samples can plausibly be explained by a concentration effect resulting from water removal during freeze-drying, or potential matrix disruption enhancing extractability during analytical preparation. These factors can lead to apparent increases in measured concentrations without necessarily reflecting net mineral enrichment.
We have now revised the discussion to provide a more balanced interpretation of these observations, acknowledging both solubility-based leaching effects and concentration phenomena related to moisture reduction and matrix alteration.
- The authors do not consider the influence of soil, fertilization, or environmental factors that may have influenced the mineral profile. For trace elements (e.g., iodine, selenium), this may be significant.
Response: We appreciate this valuable comment. However, all cherry samples were sourced from a single commercial orchard, cultivated under uniform soil, fertilization, and environmental conditions, thereby minimizing variability arising from these external factors. Consequently, the observed differences in mineral composition are most plausibly attributed to cultivar and processing effects. Nonetheless, we acknowledge that soil composition, fertilization practices, and other environmental parameters can markedly influence the trace element content of fruit. Accordingly, we have now included this consideration as a limitation in the revised manuscript conclusion.
- Although cherries contain beneficial compounds (e.g. polyphenols), direct functional data (e.g. anti-inflammatory, antioxidant effects) are lacking, so the term “health-promoting” in the conclusions may be too far-reaching without additional research.
Response: We agree with your observation and have revised the conclusion accordingly.
- In my opinion, it would be beneficial to describe the limitations of the study
Response: We have now added a new paragraph in the conclusion that describes the limitation of the study.
Round 2
Reviewer 3 Report
Comments and Suggestions for Authors
Thank you for sharing the revised manuscript. The Authors thoroughly addressed all comments and made appropriate corrections to the manuscript.